# Recombinant Cathepsin L of *Tribolium castaneum* and Its Potential in the Hydrolysis of Immunogenic Gliadin Peptides

**DOI:** 10.3390/ijms23137001

**Published:** 2022-06-23

**Authors:** Elena A. Dvoryakova, Maria A. Klimova, Tatiana R. Simonyan, Ivan A. Dombrovsky, Marina V. Serebryakova, Valeriia F. Tereshchenkova, Yakov E. Dunaevsky, Mikhail A. Belozersky, Irina Y. Filippova, Elena N. Elpidina

**Affiliations:** 1A.N. Belozersky Institute of Physico-Chemical Biology, M.V. Lomonosov Moscow State University, 119991 Moscow, Russia; greenfire06@gmail.com (E.A.D.); mserebr@mail.ru (M.V.S.); dun@belozersky.msu.ru (Y.E.D.); mbeloz@belozersky.msu.ru (M.A.B.); 2Department of Chemistry, M.V. Lomonosov Moscow State University, 119991 Moscow, Russia; maryc9-klimova@ya.ru (M.A.K.); simoniantania@yandex.ru (T.R.S.); dombrovskiy1996@yandex.ru (I.A.D.); v.tereshchenkova@gmail.com (V.F.T.); irfilipp@belozersky.msu.ru (I.Y.F.)

**Keywords:** cathepsin L, cysteine peptidases, digestive peptidases, recombinant enzymes, gliadins, *Tribolium castaneum*

## Abstract

Wheat gliadins contain a large amount of glutamine- and proline-rich peptides which are not hydrolyzed by human digestive peptidases and can cause autoimmune celiac disease and other forms of gluten intolerance in predisposed people. Peptidases that efficiently cleave such immunogenic peptides can be used in enzyme therapy. The stored product insect pest *Tribolium castaneum* efficiently hydrolyzes gliadins. The main digestive peptidase of *T. castaneum* is cathepsin L, which is from the papain C1 family with post-glutamine cleavage activity. We describe the isolation and characterization of *T. castaneum* recombinant procathepsin L (rpTcCathL1, NP_001164001), which was expressed in *Pichia pastoris* cells. The activation of the proenzyme was conducted by autocatalytic processing. The effects of pH and proenzyme concentration in the reaction mixture on the processing were studied. The mature enzyme retained high activity in the pH range from 5.0 to 9.0 and displayed high pH-stability from 4.0 to 8.0 at 20 °C. The enzyme was characterized according to electrophoretic mobility under native conditions, activity and stability at various pH values, a sensitivity to various inhibitors, and substrate specificity, and its hydrolytic effect on 8-, 10-, 26-, and 33-mer immunogenic gliadins peptides was demonstrated. Our results show that rTcCathL1 is an effective peptidase that can be used to develop a drug for the enzyme therapy of various types of gluten intolerance.

## 1. Introduction

Cathepsins are enzymes of the class of hydrolases that catalyze the hydrolysis of a peptide bond. The most extensive group consists of cathepsins containing cysteine in the active center (cysteine cathepsins). The papain C1 family, along with plant cysteine peptidases such as papain, chymopapain, and bromelain, also includes the animal lysosomal cathepsins B, C, H, F, K, L, O, S, V, W, and X/Z [1]. Lysosomal cysteine cathepsins have unique properties, are expressed in many tissues of various organisms, and play a central role in a number of important cellular processes [2]. These enzymes play an important role in antigen presentation, protein processing, and collagen hydrolysis, and they are also involved in the development of various pathologies such as rheumatoid arthritis, osteoporosis, muscular dystrophy, and Alzheimer’s disease [3].

Of particular interest are studies on insect cysteine cathepsins and the identification of a digestive function in these peptidases. Cysteine digestive peptidases are found in limited groups of insects such as infraorder Cucuiformia of the Coleoptera order [4,5,6,7,8,9,10,11] and Hemiptera order [12,13]. The digestive cysteine peptidases of stored product beetles from the family Tenebrionidae have been studied in the most detail—in particular, *Tribolium castaneum*. According to the transcriptomic data [14,15], together with biochemical [7] and proteomic [16] studies, the most highly expressed *T. castaneum* C1 endopeptidase is cathepsin L, whose gene transcripts account for 72% of the total expression level of cysteine peptidase genes in the insect larvae gut. Cathepsin L (NCBI ID NP_001164001) (TcCathL1) appears to be the main cysteine digestive peptidase in *T. castaneum* and plays an important role in the initial steps of food protein digestion [11].

Studies conducted in our laboratory have shown that the cysteine digestive peptidases of stored product pests from the Tenebrionidae family exhibit high post-glutamine hydrolyzing activity and are able to effectively hydrolyze proline- and glutamine-rich gliadins that cause a severe autoimmune hereditary celiac disease in predisposed people [17]. Celiac disease is characterized by persistent intolerance to wheat, rye, and barley prolamins and occurs in 1–2% of the world’s population. Prolamins (named gliadins in wheat—the main component of gluten) are the major storage proteins of cereals. They contain 30–50% glutamine residues and 10–30% proline residues [18]. Recent studies attempt to identify the enzymes that can effectively hydrolyze immunogenic prolamin peptides so that they can be used for the enzymatic therapy of celiac disease [19,20,21,22,23,24,25,26,27,28]. In our laboratory, the main *T. castaneum* digestive cathepsins L NP_001164001 (TcCathL1) and NP_001164314 (TcCathL2) were isolated, and their ability to cleave the fluorogenic analogues of immunogenic prolamin peptides was shown [11], which allows them to be considered as potential candidates for the treatment of celiac disease. However, the isolation of native cathepsin L from *T. castaneum* larvae is difficult due to its low stability and the complex multienzyme composition of the initial mixture. In addition, the cathepsins NP_001164001 and NP_001164314 have a 92% sequence identity, which greatly complicates the separation of these enzymes by biochemical methods. Therefore, we have obtained the main digestive cathepsin of *T. castaneum* (NP_001164001) as a recombinant protein in the yeast expression system. This paper describes the expression of this cathepsin L as a proenzyme (rpTcCathL1) and its processing to the mature enzyme, and it provides a detailed characterization of the mature enzyme properties and its ability to efficiently hydrolyze different immunogenic gliadin peptides.

## 2. Results

### 2.1. Expression of rpTcCathL1

The pUC57cat plasmid, pPICZalphaA integrative vector, and *Pichia pastoris* GS115-II-3 cells were used to express rpTcCathL1. The presence of the signal sequence of the α-factor of the yeast *Saccharomyces cerevisiae* at the N-terminal part of the translated cathepsin L allowed for the transfer of the recombinant protein directly into the culture medium. The choice of such a scheme for expression made it possible to achieve a sufficiently high yield of the target product and simplify the purification process. The genetically engineered construct did not contain a 6-His-tag, the introduction of which can adversely affect the functional properties of the target protein [29]. *T. castaneum* cathepsin L was expressed as the proenzyme (rpTcCathL1), which required subsequent processing of the inactive proenzyme into an active enzyme (rTcCathL1). A total of 100 mL of the culture supernatant contained 5.3 mg of rpTcCathL1. The resulting rpTcCathL1 was concentrated and purified from low molecular weight impurities by ammonium sulfate precipitation followed by an exhaustive dialysis against distilled water. The yield of the purified protein was 34%.

### 2.2. Autocatalytic Processing of rpTcCathL1

The processing of rpTcCathL1 was carried out autocatalytically. In order to optimize this process, the effects of pH and proenzyme concentration in the medium during processing on the rate of processing were studied.

#### 2.2.1. Effect of pH

The study of the autocatalytic processing of rpTcCathL1 was carried out in the pH range from 3.0 to 7.0 at 37 °C in the processing medium (Figure 1). The optimal pH values for the process were pH 4.0, 4.5, and 5.0. At these pH values, the enzyme activity increased significantly, reaching similar maximum values in 40, 60, and 75 min, respectively. This value was stable at least until 180 min, forming a processing plateau. At the pH values of 3.0 and 3.5, an increase in enzyme activity was observed, but no processing plateau was formed, and the activity rapidly decreased. At a pH value of 5.5, an increase in activity and the formation of a processing plateau were also observed; however, the rate of processing and the maximum activity were 2–3 times lower than those during processing under more acidic conditions. At more alkaline pH values from 6.0 to 7.0, almost no conversion of rpTcCathL1 was observed within 180 min of the experiment, as evidenced by the lack of pronounced activity in the hydrolysis of the chromogenic substrate pyroglutamyl-phenylanyl-glutamyl-*p*-nitroanilide (Glp-Phe-Gln-pNA) during this period of time. Based on the data on the maximum processed activity, the time to reach it, and the duration of the processing plateau, the optimal conditions for the processing of rpTcCathL1 preparation to the rTcCathL1 used for further studies were incubation at pH 4.0 and 37 °C.

#### 2.2.2. Effect of Proenzyme Concentration on the Processing Rate

The effect of proenzyme concentration on the processing of rpTcCathL1 was studied at proenzyme concentrations from 0.2 to 5.0 mg/mL at pH 4.0 (Figure 2). Activity measurements of the processed preparation were performed at the same concentration of rTcCathL1 equal to 0.004 mg/mL with the Glp-Phe-Gln-pNA substrate. During the experiment, the minimal time required to reach the processing plateau (maximum enzyme activity) was determined for each concentration. The results obtained indicate that the time required for the enzyme activity to reach a plateau increases with an increase in the proenzyme concentration in the reaction mixture, that is, the processing rate decreases. At a concentration of rpTcCathL1 of 0.2 mg/mL, 35 min pass from the beginning of processing to reaching a plateau, and at a concentration of rpTcCathL1 of 5 mg/mL, this process takes 98 min. That is why, in the subsequent experiments, we used an enzyme concentration of no more than 0.4 mg/mL.

### 2.3. Electrophoretic Study of rpTcCathL1 Processing

The progress of processing was analyzed using native PAGE combined with in-gel post-electrophoretic activity testing with the fluorogenic substrate Glp-Phe-Ala-AMC (where AMC is 4-amino-7-methylcoumaride) (Figure 3A,B). Post-electrophoretic activity testing revealed an increase in the activity of the enzyme with the fluorogenic substrate upon the cleavage of the propeptide during the autocatalytic processing of rpTcCathL1. It can be seen that a small amount of the active enzyme is present in the mixture from the very beginning, which is confirmed by the presence of a fluorescent band in the gel. The Coomassie-stained panel of the processed protein (Figure 3B) corresponds to the activity panel (Figure 3A), as confirmed by the coincidence of Rf factors. After 40 min of incubation, it can be observed that the major part of the proenzyme has been processed, and after 60 min of processing, the proenzyme is no longer present in the reaction mixture.

### 2.4. Mass-Spectrometry Analysis of rpTcCathL1 and rTcCathL1

For the mass-spectrometry (MS) analysis of rpTcCathL1 and rTcCathL1, we used hydrolysates obtained with glutamate-specific peptidase and trypsin (Figure 4). For MS analysis of the mature enzyme, a sample was taken corresponding to 60 min of processing (Figure 3). The results show that the sequence of the rpTcCathL1 really corresponds to the sequence of procathepsin L *T. castaneum* NP_001164001. The sample corresponding to the mature enzyme did not contain peptides corresponding to the propeptide part of the protein. The use of glutamate-specific peptidase made it possible to identify a part of the amino acid sequence of the protein from residue 157 to 255, containing only one arginine residue, which was not hydrolyzed by trypsin.

The results of the MS analysis of mature rTcCathL1 after trypsinolysis showed the presence of several peptides at the putative N-terminus of the enzyme differing by several amino acids

SGKPAAAEVDWR

 GKPAAAEVDWR

  KPAAAEVDWR

   AAEVDWR

Apparently, several forms of the mature enzyme are present in the final preparation of the mature rTcCathL1. The shortest peptide coincides with the found N-terminus of the native cathepsin L *T. castaneum* [11].

### 2.5. Effect of pH on the Activity and Stability of rTcCathL1

The pH-optimum for rTcCathL1 was found at pH 8.0, and the enzyme retained high activity in a wide pH range from 5.0 to 9.0 (64–94% of the maximum activity level) (Figure 5). The pH-stability was tested at 20 °C and 37 °C. Upon the incubation of rTcCathL1 at 20 °C, a wide interval (4.0–8.0) of high pH-stability (76–90% of the maximum activity level) was tested (Figure 6A). At pH values below 3.0 and above 9.0, the stability of the preparation sharply reduced (1–15% of the maximum level of activity). The analysis at 37 °C showed that the range of pH-stability of rTcCathL1 sharply narrowed and shifted to the acidic region; the area of high stability (75–90% of the maximum activity level) was observed in the pH range of 4.0–6.0 (Figure 6B). At pH values below 4.0 and above 6.0, the stability of the preparation was sharply reduced.

### 2.6. Effect of Inhibitors on the Activity of rTcCathL1

The effect of inhibitors on the activity of rTcCathL1 was tested using the general inhibitor of the cysteine peptidase N-(trans-Epoxysuccinyl)-L-leucine 4-guanidinobutylamide (E-64) and the selective inhibitors of the cysteine cathepsins L II, III, and IV (Figure 7). The results of IC_50_ comparison show that the best inhibitor was the general inhibitor of the cysteine peptidase E-64, followed by the selective inhibitors of the cathepsins L II and III, which inhibit it at nanomolar concentrations.

### 2.7. Hydrolysis of Chromogenic Substrates by rTcCathL1

The substrate specificity of rTcCathL1 was studied on a series of selective chromogenic *p*-nitroanilide substrates (Glp-Phe-Gln-pNA, Glp-Phe-Leu-pNA, Glp-Phe-Ala-pNA, and Glp-Val-Ala-pNA) designed and synthesized in our laboratory and the commercially available arginine-containing substrate Z-Phe-Arg-pNA, where Z is benzyloxycarbonyl, and Z-Arg-Arg-pNA is hydrolyzed by cysteine as well as all trypsin-like peptidases (Figure 8). The Z-Phe-Arg-pNA substrate, which satisfies the classical requirements for cysteine peptidase substrates as well as trypsins, was the most preferred for rTcCathL1. The efficiency of the substrate hydrolysis decreased in the series Z-Phe-Arg-pNA > Glp-Phe-Gln-pNA > Glp-Phe-Leu-pNA > Glp-Phe-Ala-pNA > Glp-Val-Ala-pNA > Z-Arg-Arg-pNA.

### 2.8. Hydrolysis of Toxic Gliadin Peptides

The ability of rTcCathL1 to hydrolyze toxic gliadin peptides was studied using the hydrolysis of four immunogenic peptides: 8-mer γ-5-gliadin peptide QPQQPFPQ, 10-mer α-2-gliadin peptide LPYPQPQLPQ, 26-mer γ-5-gliadin peptide FLQPQQPFPQQPQQPYPQQPQQPFPQ, and 33-mer α-2-gliadin peptide LQLQPFPQPQLPYPQPQLPYPQPQLPYPQPQPF. The sites of hydrolysis were determined by MS analysis.

The mass spectrum of the original 8-mer peptide QPQQPFPQ (Figure 9A) contained a peak with a value of 968.5 *m*/*z* belonging to the original peptide and several impurity peaks, presumably by-products of peptide synthesis, with values of 951.5 *m*/*z* and 973.8 *m*/*z*. The mass spectrum of the hydrolysis products showed only one peak corresponding to the QPFPQ peptide formed during the QPQ↓QPFPQ cleavage at the Q–Q bond (Figure 9B). The second QPQ product had a low molecular mass (371.4 kDa), which is below the resolution limit of this method. The peak of the original 8-mer peptide and the accompanying impurity peaks were absent in Figure 9B.

The mass spectrum of the original 10-mer LPYPQPQLPQ peptide contained a peak with an *m*/*z* value of 1179.7, corresponding to the original peptide (Figure 10A). In the spectrum of the hydrolysis products of this peptide, there was a single peak of the hydrolysis product at the Q–L bond (LPYPQPQ↓LPQ): the LPYPQPQ peptide with an *m*/*z* value of 842.5 (Figure 10B). The second peak of the product LPQ (356.4 kDa) is also below the resolution limit of the method. There was no peak of the original peptide.

The mass spectrum of the original 26-mer FLQPQQPFPQQPQQPYPQQPQQPFPQ peptide contained a peak with an *m*/*z* value of 3146.6, corresponding to the original peptide (Figure 11A). In the spectrum of hydrolysis products, there were three major peaks of the hydrolysis at the Q–Q bonds in the formula: FLQPQ↓QPFPQ↓QPQ↓QPYPQ↓QPQ↓QPFPQ (Figure 11B). We can see a peak of the QPFPQ peptide with an *m*/*z* value 616.3, which is the final product of the hydrolysis of this peptide. Two remaining peaks belong to the peptides QPQQPFPQ (*m*/*z* 969.5) and QPYPQQPQ (*m*/*z* 985.5), which are the products of partial hydrolysis. Peptide QPQ (371.4 kDa), probably formed during hydrolysis, is below the resolution limit of the method.

The mass spectrum of the original 33-mer LQLQPFPQPQLPYPQPQLPYPQPQLPYPQPQPF peptide contained a peak with an m/z value of 3910.9, corresponding to the original peptide (Figure 12A). In the spectrum of hydrolysis products of this peptide, there were six major peaks of the hydrolysis products of the Q-L bonds, (LQ↓LQPFPQPQ↓LPYPQPQ↓LPYPQPQ↓LPYPQPQPF) (Figure 12B, Table 1). The mass spectrum contains peaks of the LPYPQPQ peptide with an *m*/*z* value of 842.5, the LQPFPQPQ peptide with an *m*/*z* value 954.5, and the LPYPQPQPF peptide with an *m*/*z* value 1086.6, which are the final products of the hydrolysis of this peptide. The remaining peaks belong to the peptides LPYPQPQLPYPQPQ, LQPFPQPQLPYPQPQ, and LPYPQPQLPYPQPQPF, which are the products of partial hydrolysis. The peptide LQ (259.3 kDa) is below the resolution limit of the method.

## 3. Discussion

Cysteine proteolytic enzymes play an important role in the degradation of proteins in lysosomes and endosomes. They also participate in the processes of cell proliferation, invasion, and metastasis, ensuring the degradation of the extracellular matrix and the destruction of intercellular interactions [2]. Another subgroup of extracellular cysteine cathepsins play an important role in digestion in several groups of insects including the tenebrionid family due to their ability to degraderesistant to hydrolysis food proteins [17], which include gluten proteins (gliadins) of wheat as well as prolamins from rye and barley that are rich in glutamine and proline. Such structural features of gliadins determine the high resistance of some peptides of these proteins to the action of human digestive peptidases, and it is these immunogenic peptides that cause celiac disease in predisposed people [30]. Patients with celiac disease suffer from allergic reactions in the form of atrophy of the intestinal mucosa, malabsorption, weight loss, and in the form of an increased risk of developing osteoporosis and T-cell lymphomas [31]. At the moment, the only available treatment strategy for celiac disease patients is a lifelong gluten-free diet, but even with the strict control of gluten-free products, the prolamin levels can sometimes be higher than declared due to the phenomenon of cross-contamination or contamination during the food production process [32]. An alternative to such a diet for predisposed people can be enzyme therapy using proteolytic enzymes capable of cleaving immunogenic peptides. A convenient source of such enzymes can be digestive enzymes of representatives of the Tenebrionidae family, whose main diet includes gluten proteins rich in proline and glutamine, particularly the stored product insect pest *T. castaneum*. It should be noted that, whereas proline-specific peptidases that cleave bonds formed by proline are relatively well studied [19,20], only a limited range of glutamine-specific peptidases are known [28]. Therefore, we evaluated the activity of *T. castaneum* cathepsin L as a prospective candidate for the development of an effective pharmaceutical for the therapy of celiac disease and other gluten intolerances.

Taking into account the low stability of native cathepsin L isolated directly from *T. castaneum* larvae and the technical complexity of obtaining a pure individual enzyme, we developed a procedure for obtaining a recombinant proenzyme rpTcCathL1 using the yeast expression system. The comparison of various expression systems indicates that the yeast *Pichia pastoris* system is convenient for obtaining active recombinant cysteine cathepsins of the C1 family since it allows for the formation of regular disulfide bonds and the obtaining of the target product in a high yield in a culture medium with post-translational modifications, if necessary [33]. In order to avoid autolysis, we chose the proenzyme as the initial form since it does not have enzymatic activity and is easily processed into the active form.

In living cells, lysosomal procathepsin L undergoes proteolytic processing into the active, mature form of cathepsin in the acidic environment of late endosomes or lysosomes [34]. Digestive procathepsins L are presumably processed in the acidic gut tissue vesicles or acidic gut contents [8]. In the larvae guts of the Tenebrionidae family, pH values increase from 5.2 in the anterior midgut part to 8.2 in the posterior part, which apparently promotes the processing of cathepsin L in the anterior midgut, determining its initial role in insect digestion [4,7,8]. We showed that pH values of 4–5 are optimal for rpTcCathL1 processing; the enzyme reaches its maximum activity and the processing plateau in 40–75 min at these pH values and at an enzyme concentration of 0.4 mg/mL. The formation of a processing plateau, where the activity of the enzyme does not change, and electrophoretic data suggest that, during this period, the entire rpTcCathL1 is processed into the mature form rTcCathL1. At pH 5.5, the processing rate is reduced, and at more alkaline pH values, processing practically does not occur. At pH values of 3 and 3.5, the enzyme activity reached its maximum values faster—in 15 and 30 min, respectively—but after that, it began to decrease rapidly, which apparently was associated with rTcCathL1 inactivation.

Studying the effect of rpTcCathL1 concentration in the reaction mixture on the processing rate, we showed that, with an increase in the proenzyme concentration in the solution, the time required for the enzyme activity to reach a plateau increased. This may be due to the fact that, during the processing of increasing concentrations of rpTcCathL1, the accumulation of the cleaved propeptide slows down the further course of the reaction [35] since the propeptide is an inhibitor of cathepsins L [2].

The study of rpTcCathL1 processing by native PAGE showed a gradual accumulation of the processing product over time and the disappearance of the initial band. An increase in enzymatic activity in the product band confirms that mature rTcCathL1 is accumulating. The low enzymatic activity observed at zero processing time is apparently due to the presence of a small amount of autoprocessed mature cathepsin in the proenzyme preparation. No intermediate forms of processed rpTcCathL1 could be found on the electrophoregram.

The identity of the obtained rpTcCathL1 was confirmed by MS analysis. It corresponded to the amino acid sequence of the main digestive cathepsin L of *T. castaneum* [11]. Using MS/MS analysis, we obtained a number of peptides located at the N-terminus of the processed rpTcCathL1, which differ by several amino acids. The shortest of them coincided with the N-terminus of the native cathepsin L. This suggests that the processing is either ambiguous or multi-stage. The analysis of the in vitro processing of the orthologous major digestive cathepsin L of the closely related tenebrionid *Tenebrio molitor* also revealed two amino acid extensions of the N-terminal peptide in the mature form [8]. The presence of 1–6 additional residues at the N-terminus of the mature enzyme has also been shown in a number of studies on the processing of mammalian [35,36] and helminth cathepsins L [37]. It may be concluded that rpTcCathL1 processing occurs either step by step, with sequential amino acid cleavage at the N-terminus, or with the formation of several active forms of the mature enzyme.

rTcCathL1 was active over a wide pH range from 3 to 10. Its stability was studied at two temperatures, 20 °C and 37 °C, which correspond to the temperatures in the insect and human organisms. At lower temperatures, rTcCathL1 displayed a high stability over a pH range of 4 to 8. With an increase in the incubation temperature to 37 °C, the maximum stability of rTcCathL1 shifted to the acidic region; at pH 7, the activity significantly decreased, which corresponds to the data in the literature [38]. According to these data, this effect can apparently be explained by the irreversible unfolding of cathepsin L under these conditions.

The results of the inhibitory analysis showed that the most effective rTcCathL1 inhibitor is the universal inhibitor E-64.

Among the selective substrates containing the Phe residue in the P2 position, rTcCathL1 most actively hydrolyzed Glp-Phe-Gln-pNA, followed by Glp-Phe-Leu-pNA and Glp-Phe-Ala-pNA, so it can be assumed that the glutamine residue in the P1 position is the most preferred, and the leucine residue in this position is preferred over alanine. The Glp-Phe-Ala-pNA and Glp-Val-Ala-pNA substrates were hydrolyzed approximately at the same rate, which indicates a similar effect of the Phe and Val residues in the P2 position on the hydrolysis efficiency. Among the studied commercial substrates, the universal substrate of cysteine and the trypsin-like peptidase Z-Phe-Arg-pNA was hydrolyzed slightly better than Glp-Phe-Gln-pNA, whereas Z-Arg-Arg-pNA was hydrolyzed with the least efficiency.

Testing the ability of rTcCathL1 to hydrolyze toxic gliadin peptides showed that the enzyme was able to efficiently cleave these peptides, including the highest molecular mass 26-mer and 33-mer peptides, hydrolyzing them at the Q–Q and Q–L bonds. The 8-mer QPQQPFPQ peptide is hydrolyzed at the Q–Q bond to form five- and three-membered reaction products, whereas the 10-mer LPYPQPQLPQ is cleaved at the Q–L bond to form the products LPYPQPQ and LPQ. The 26-mer FLQPQQPFPQQPQQPYPQQPQQPFPQ peptide is hydrolyzed at the Q–Q bond, but the hydrolysis is not complete. The mass spectrum contains both peptides that are the products of complete hydrolysis and those of partial hydrolysis. In the hydrolysis spectrum of the 33-mer LQLQPFPQPQLPYPQPQLPYPQPQLPYPQPQPF peptide, the products of the complete and partial hydrolysis of this peptide at the Q–L bond are also observed. However, none of the mass spectra of the hydrolysates contain the peak of the original peptide; therefore, it can be assumed that, with an increase in the hydrolysis time or the amount of rTcCathL1, the intermediate hydrolysis products will be completely cleaved. The peptides QPQ (in the first case), LPQ (in the second case), and LQ (in the fourth case) are not identified by MALDI-TOF MS due to their low molecular mass.

Thus, the totality of the data obtained indicates that rTcCathL1 is an effective peptidase capable of hydrolyzing immunogenic gliadin peptides and can be used to develop a drug for the enzyme therapy of various types of gluten intolerance, as well as for processing raw materials or preparing gluten-free food products from wheat, rye, and barley. The major digestive peptidase of *Tribolium castaneum*, the recombinant preparation of which is rTcCathL1, is purposefully adapted for prolamin hydrolysis, and this adaptation can provide greater efficiency in the cleavage of gliadins and their toxic peptides as compared to the analogs from the heterologic systems.

## 4. Materials and Methods

### 4.1. Cloning and Expression of rpTcCathL1

The cloning of the TcCathL1 gene into the pUC57cat plasmid was performed in GenScript (Piscataway, NJ, USA) by standard procedures. The 960 bp procathepsin L gene was cloned from the pUC57cat plasmid into the pPICZalphaA integrative vector. To amplify the cathepsin gene, primers were used to the ends of the gene, pUC57cat was used as a template, and High Fidelity Enzyme Mix polymerase was obtained from Thermo Fisher Scientific Inc (Waltham, MA, USA). The DNA fragment obtained by PCR was purified by agarose gel electrophoresis and isolated using the DNA extraction Kit from Thermo Fisher Scientific Inc. (Waltham, MA, USA). The PCR fragments were flanked by XbaI and XhoI restriction sites. After treatment with the restrictases XbaI and XhoI, the pTcCathL1 gene was cloned into the pPICZalphaA vector. The gene was sequenced to eliminate PCR errors. In the resulting recombinant plasmid, the procathepsin L gene is under the control of the AOX1 promoter and is fused in the same reading frame with the signal sequence of the α-factor of the yeast *S. cerevisiae*, which provided its secretion into the culture medium.

To construct the rpTcCathL1 producer, the recombinant plasmid was linearized and introduced into *P. pastoris* GS115-II-3 cells by electroporation. The selection of transformants was carried out on plates with the YPD agar medium containing zeocin for selection. The selection of clones with the rpTcCathL1 gene integrated into the chromosome was carried out using PCR. The biomass from the Petri plates was collected and suspended in sterile distilled water. The inoculation of baffled shake flasks containing 100 mL of the BMMY fermentation medium was carried out with one-unit OD_600_/_mL_. To induce pTcCathL1 gene expression, methanol was added to the medium up to 1% (*v*/*v*) every 24 h. Fermentation was carried out on a New Brunswick Innova 43 shaker at 28 °C and 250 rpm for 65 h. After fermentation, the culture medium containing rpTcCathL1 was separated from the cells by centrifugation. Aliquots of the culture supernatant were stored at −75 °C.

### 4.2. Isolation of rpTcCathL1

Dry ammonium sulfate was added with stirring to a certain volume of the culture medium in an amount of 80% by weight and left at 4 °C overnight. The resulting precipitate was separated by centrifugation at 8000 rpm for 15 min and dissolved in 0.5 mL of distilled water. The precipitate was dialyzed against distilled water in a dialysis bag for 12 h. The dialysate obtained was stored at −75 °C.

### 4.3. Protein Concentration Assay

The amount of protein in the sample was determined by the Bradford method [39]. A total of 5 μL of the enzyme was added to a microplate well, 150 µL of the Bradford reagent was added, and the absorbance of the solution was measured at 595 nm using an ELx808 Absorbance Microplate Reader (Agilent, Santa Clara, CA, USA). The protein concentration in the initial preparation in mg/mL was determined from the calibration curve constructed using the stock solutions of BSA at different concentrations.

Additionally, the protein concentration was measured by the optical absorption of the protein solution at λ = 280 nm in a quartz cuvette with l = 1 cm on a Genesys 10 S UV spectrophotometer (Thermo Fisher Scientific Inc., Waltham, MA, USA). The protein concentration was calculated using the molar extinction coefficient 65,080 M^−1^ ×* cm^−1^ for rpTcCathL1 and 55,550 M^−1^ × cm^−1^ for rTcCathL1, as obtained by the GPMAW program (Lighthouse data, Odense, Denmark).

### 4.4. Assay of Peptidase Activity with the Chromogenic Substrate Glp-Phe-Gln-pNA

The enzymatic activity was measured in a 96-well plate using an ELx808 Absorbance Microplate Reader (Agilent, Santa Clara, CA, USA) at 405 nm according to [11]. The measurements were performed in the presence of 6 mM cysteine and 1 mM EDTA in 0.1 M acetate–phosphate–borate universal buffer (UB), pH 5.6 [40] using a selective substrate 0.5 mM Glp-Phe-Gln-pNA [41]. The reaction mixture was incubated at 37 °C. The activity was calculated in nmol/min from the initial rate of hydrolysis.

### 4.5. Processing of rpTcCathL1

The *T. castaneum* rpTcCathL1, prepared by precipitation with ammonium sulfate followed by dialysis, was diluted with 0.1 M acetate buffer pH 4.0 to the required enzyme concentration, the cysteine was added to a concentration of 6 mM, and it was incubated at 37 °C. Samples of the reaction mixture were taken every 10 min, and the activity was measured according to the method described in paragraph 4.4 using the Glp-Phe-Gln-pNA substrate. The processed preparation was stored at −75 °C.

### 4.6. Effect of pH on the rpTcCathL1 Processing Rate

A total of 54 µL of 0.1 M UB with pH values in the range of 3.0–7.0 with a pH value step of 0.5 and 4 µL of 120 mM cysteine solution were added to the 12 µL of rpTcCathL1 preparation at a concentration of 5.7 mg/mL. Thus, the final concentration of rpTcCathL1 in the processing mixture was 0.4 mg/mL, and the concentration of cysteine was 6 mM. The experiment was carried out for 180 min at 37 °C, taking samples from the processing mixture every 10 min for the first 60 min of the experiment and every 30 min for the remaining 120 min of the experiment. Prior to the activity measurement, the sample was further diluted with 0.1 M UB, pH 5.6 to obtain a 0.004 mg/mL concentration of processed rpTcCathL1 (rTcCathL1), and the rTcCathL1 activity was measured according to the procedure described in Section 4.4.

### 4.7. Effect of the Proenzyme Concentration on the rpTcCathL1 Processing Rate

The effect of the proenzyme concentration on the rpTcCathL1 processing rate was studied with the final concentrations of the proenzyme in the reaction mixture from 0.2 to 5.0 mg/mL at 37 °C. The reaction mixture contained 0.1 M UB, pH 4.0, and 6 mM cysteine. Each concentration of rpTcCathL1 was processed according to the same time scheme as that described in Section 4.4. The processed samples of rTcCathL1 were diluted with 0.1 M UB, pH 5.6 to an enzyme concentration of 0.004 mg/mL, and the activity of rTcCathL1 was measured according to the method described in 4.4. The rate of processing was recorded as the minimal time required for the activity of a mature enzyme to reach a plateau with a maximum activity value.

### 4.8. Native PAGE and Post-Electrophoretic Detection of Proteolytic Activity

Native PAGE of the rpTcCathL1 processing products was performed in the direction from the cathode to the anode in a 12% separating and 4% concentrating polyacrylamide gel in a buffer containing 35 mM 4-(2-hydroxyethyl)-1-piperazineethanesulfonic acid (HEPES) and 43 mM imidazole, at pH 7.2, according to McLellan [42]. In-gel post-electrophoretic testing of the activity was performed using 75 µM fluorogenic substrate Glp-Phe-Ala-AMC [43] according to [44]. The localization of proteolytic activity bands was identified under a UV lamp at 366 nm.

After proteolytic activity testing with a fluorogenic substrate, to visualize the protein bands, the gel block was placed in a 0.1% Coomassie G-250 solution in a mixture of ethanol/acetic acid/water (3:1:6 *v*/*v*/*v*), washed with 7% acetic acid solution, and photographed.

### 4.9. Mass Spectrometry Analysis of rpTcCathL1 and rTcCathL1

The stained bands after native electrophoresis corresponding to the proenzyme protein band (0 min of processing) and active mature enzyme after 60 min of processing were excised from the gel and subjected to in-gel trypsinolysis or hydrolysis by the glutamate-specific peptidase V8 of *Staphylococcus aureus* [45]. Mass spectrometric (MS) analysis and MS/MS analysis of the obtained hydrolysates were carried out on a tandem Ultraflex MALDI-TOF-TOF mass spectrometer (Bruker Daltonik, Bremen, Germany) equipped with a UV laser (Nd). The accuracy of the measured masses after additional calibration with trypsin autolysis peaks was 30 ppm (0.01%). Fragmentation spectra were obtained in the tandem LIFT mode, the accuracy of measurement of the fragment mass was 1 Da. The resulting mass spectra were analyzed using the Mascot Server [46] (http://www.matrixscience.com, 16 May 2022) and GPMAW program (Lighthouse data, Odense, Denmark).

### 4.10. Effect of pH on rTcCathL1 Activity and Stability

The effect of pH on rTcCathL1 activity was studied in 0.1 M UB with pH values from 2 to 10 with a step of 1 pH unit in the presence of 6 mM cysteine and 1 mM EDTA. Activity measurements were performed using the Glp-Phe-Gln-pNA substrate, as described in 4.4.

The pH-stability of the enzyme was tested during 2 h incubations in 0.1 M UB, with pH values ranging from 1.9 to 10.7 at 20 °C and 37 °C. After 2 h, the pH of all the samples was adjusted to a value of 5.6, and the activity was measured as described above.

### 4.11. Substrate Specificity of rTcCathL1

Substrate specificity was studied using the 0.25 mM chromogenic substrates Z-Phe-Arg-pNA (where Z = benzyloxycarbonyl), Z-Arg-Arg-pNA (both from Bachem, Switzerland), Glp-Phe-Gln-pNA, Glp-Phe-Leu-pNA, Glp-Phe-Ala-pNA, and Glp-Val-Ala-pNA, as described in Section 4.4. The substrates Glp-Phe-Leu-pNA, Glp-Val-Ala-pNA, Glp-Phe-Ala-pNA, Glp-Phe-Ala-AMC, and Glp-Phe-Gln-pNA were synthesized, as described in [41,43,47].

### 4.12. Effect of Inhibitors on Cysteine Cathepsin

Inhibitor sensitivity was studied using E-64 (Bachem AG, Bubendorf, Switzerland) (10^−8^ or 10^−9^ M in 50% ethanol), the cathepsin L II inhibitor (Z-Phe-Tyr-CHO) (10^−5^, 10^−6^, 10^−7^ or 10^−8^ M), the cathepsin L III inhibitor (Z-Phe-Tyr(t-Bu)-DMK (where DMK = diazomethylketone) (10^−5^, 10^−6^, 10^−7^ or 10^−8^ M), and the cathepsin inhibitor L IV (1-naphthalenesulfonyl-Ile-Trp-CHO) (10^−5^, 10^−6^, 10^−7^, or 10^−8^ M) (all Calbiochem, San Diego, CA, USA). The stock solution of E-64 was prepared in 50% ethanol, and the stock solutions of the other inhibitors were prepared in 100% dimethyl sulfoxide. The final molar concentration of rTcCathL1 was 3.0 × 10^−8^ M. The enzyme was incubated in the presence of various concentrations of the studied inhibitors at room temperature for 60 min. The residual activity was measured using the Glp-Phe-Gln-pNA substrate according to the procedure described in Section 4.4. The inhibitors were compared by the concentration that resulted in a 50% inhibition of proteolytic activity (IC_50_). The calculations were performed by the least squares’ method.

### 4.13. Study of Immunogenic Peptides Hydrolysis

The hydrolysis of the immunogenic peptides was studied using 10 mM stock solutions in dimethylformamide (DMF) of QPQQPFPQ (8-mer), LPYPQPQLPQ (10-mer), FLQPQQPFPQQPQQPYPQQPQQPFPQ (26-mer), and LQLQPFPQPQLPYPQPQLPYPQPQLPYPQPQPF (33-mer). The reaction mixture included 0.25 μM rTcCathL1, 6 mM cysteine, and 0.25 mM peptide in 0.02 M ammonium acetate buffer, pH 5.6, and the reaction mixture volume was 200 mL. The final enzyme:peptide ratio was [E]:[S] = 1:1000. The hydrolysis was carried out at 37 °C for 40 min and was stopped by adding 5 μL of concentrated acetic acid. The results of the hydrolysis were visualized by a MALDI-TOF MS analysis of the hydrolysates. The reaction mixture containing no enzyme was used as a control.

### 4.14. Statistics

All of the assays were made in three replicates in three or four independent experiments. The specific enzyme activities were calculated using the initial rates of hydrolysis. The enzyme activity values were calculated as the mean of all the independent experiments. The errors were calculated as standard deviations. The calculations of the activity values were performed by nonlinear regression using the program OriginPro 8.0 (OriginLab, Northampton, MA, USA).

## Figures and Tables

**Figure 1 ijms-23-07001-f001:**
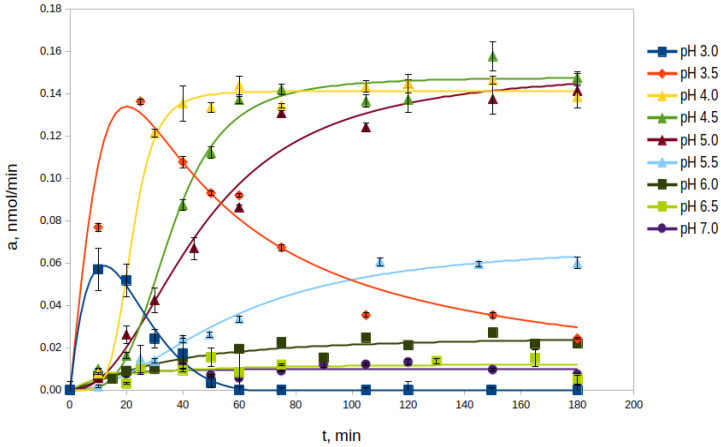
Effect of pH on the rate of rpTcCathL1 processing. The activity of the mature rTcCathL1 was measured with 0.5 mM Glp-Phe-Gln-pNA in 0.1 M universal buffer (UB), pH 5.6 in the presence of 6 mM cysteine and 1 mM ethylenediaminetetraacetic acid (EDTA).

**Figure 2 ijms-23-07001-f002:**
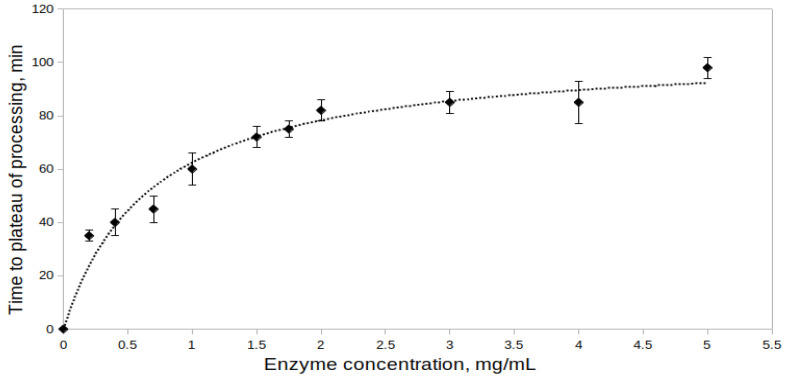
Effect of proenzyme concentration in the processing reaction mixture on the processing of rpTcCathL1. The activity of the mature rTcCathL1 was measured with 0.5 mM Glp-Phe-Gln-pNA in 0.1 M UB, pH 5.6 in the presence of 6 mM cysteine and 1 mM EDTA.

**Figure 3 ijms-23-07001-f003:**
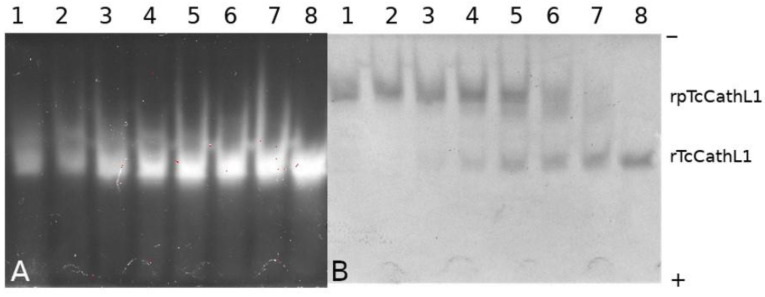
Native PAGE of rpTcCathL1 processing products during incubation from 0 to 60 min; the proenzyme concentration was 0.12 mg/mL. (**A**) Lanes 1–8: post-electrophoretic activity testing with 75 µM fluorogenic substrate Glp-Phe-Ala-AMC of rpTcCathL1 processing products after 0, 5, 10, 15, 20, 30, 40, and 60 min of processing; (**B**) lanes 1–8 rpTcCathL1 processing products after 0, 5, 10, 15, 20, 30, 40, and 60 min of processing stained with Coomassie G-250 after activity testing.

**Figure 4 ijms-23-07001-f004:**
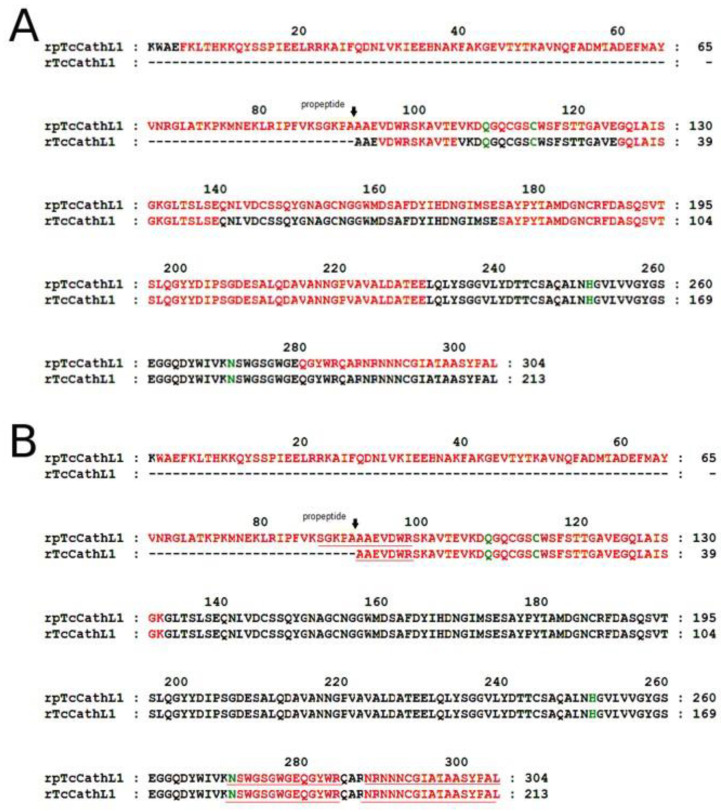
MS analysis of rpTcCathL1 and rTcCathL1. Peptides identified by peptide fingerprinting using matrix-assisted laser desorption/ionization time of flight mass spectrometry (MALDI-TOF MS) are marked in red; peptides sequenced by MS/MS analysis are underlined; amino acid residues of the active center are marked in green. The proposed start of the mature enzyme is indicated by a black arrow. (**A**) Hydrolysis by glutamate-specific peptidase, (**B**) hydrolysis by trypsin.

**Figure 5 ijms-23-07001-f005:**
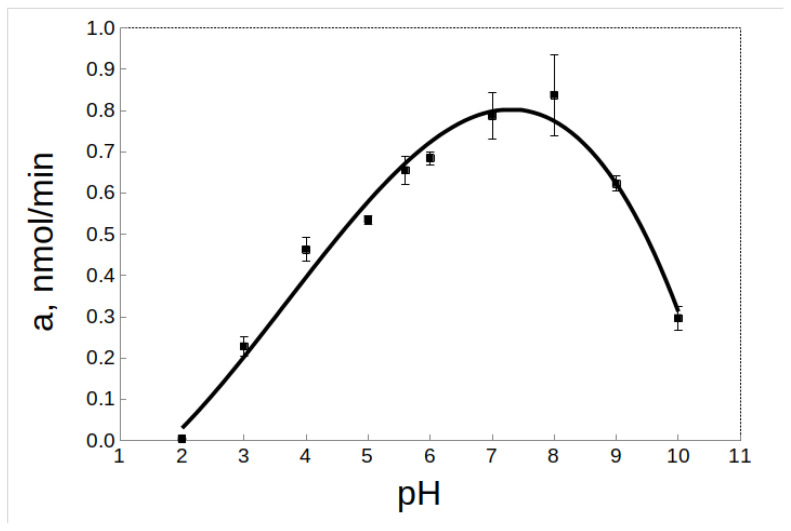
Effect of pH on the activity of rTcCathL1 with the substrate Glp-Phe-Gln-pNA in 0.1 M UB, pH 5.6 in the presence of 6 mM cysteine and 1 mM EDTA.

**Figure 6 ijms-23-07001-f006:**
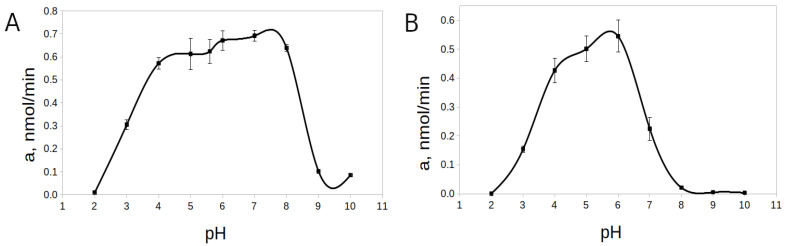
pH-stability of rTcCathL1 at 20 °C (**A**) and 37 °C (**B**). Activity was measured with the substrate Glp-Phe-Gln-pNA in 0.1 M UB, pH 5.6 in the presence of 6 mM cysteine and 1 mM EDTA.

**Figure 7 ijms-23-07001-f007:**
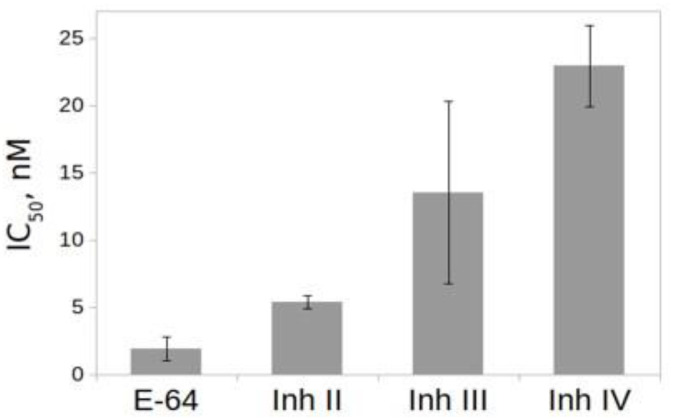
Effect of inhibitors on the activity of rTcCathL1. IC_50_ (nM) values were measured with 0.5 mM Glp-Phe-Gln-pNA in 0.1 M UB, pH 5.6 in the presence of 6 mM cysteine. Cathepsin L inhibitors II, III, and IV are named Inh II, III, and IV, respectively.

**Figure 8 ijms-23-07001-f008:**
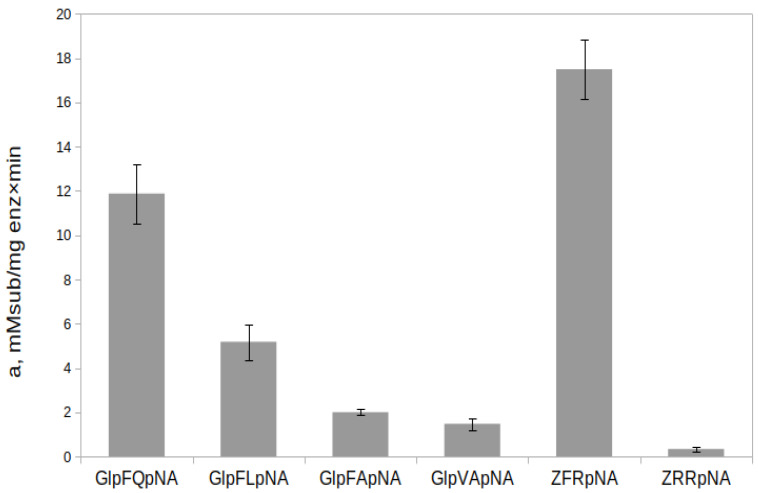
Substrate specificity of rTcCathL1 measured for a range of dipeptide substrates. Activity was measured in 0.1 M UB, pH 5.6 in the presence of 6 mM cysteine and 1 mM EDTA.

**Figure 9 ijms-23-07001-f009:**
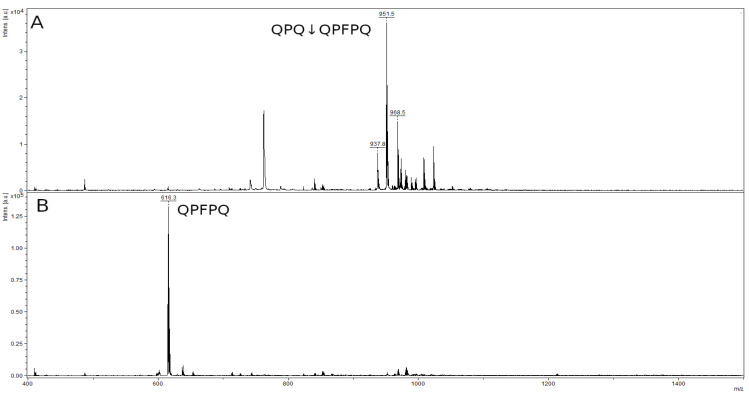
Mass spectra of the original 8-mer peptide QPQQPFPQ (**A**) and hydrolysate by rTcCathL1 after 40 min of incubation (**B**).

**Figure 10 ijms-23-07001-f010:**
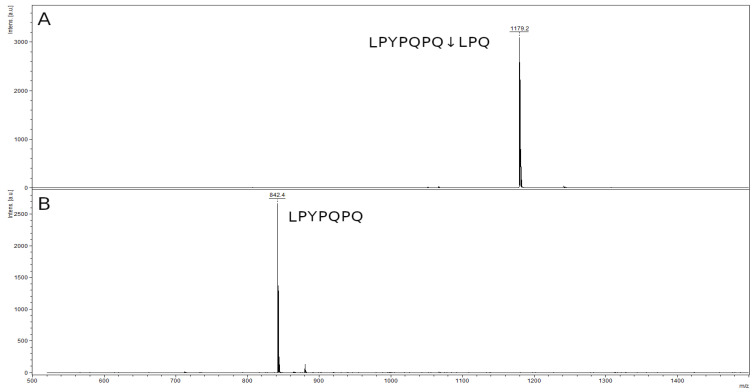
Mass spectra of the original 10-mer LPYPQPQLPQ peptide (**A**) and hydrolysate by rTcCathL1 after 40 min of incubation (**B**).

**Figure 11 ijms-23-07001-f011:**
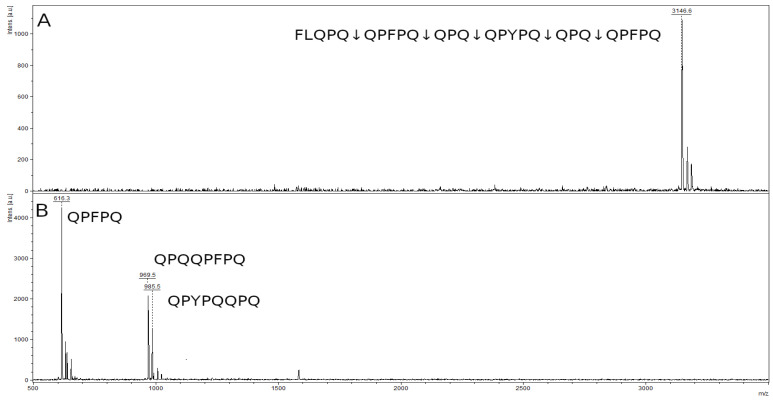
Mass spectra of the original 26-mer FLQPQQPFPQQPQQPYPQQPQQPFPQ peptide (**A**) and hydrolysate by rTcCathL1 after 40 min of incubation (**B**).

**Figure 12 ijms-23-07001-f012:**
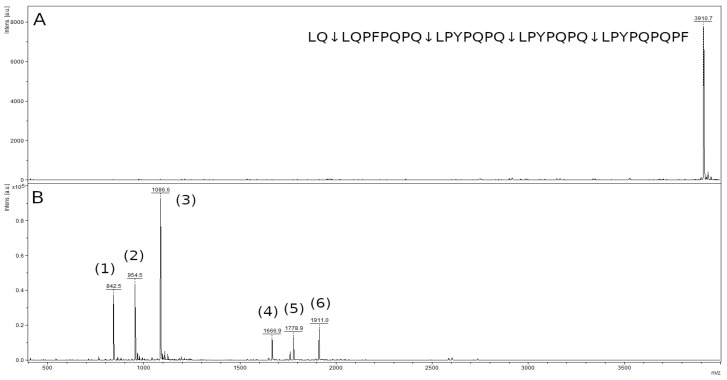
Mass spectra of the original 33-mer LQLQPFPQPQLPYPQPQLPYPQPQLPYPQPQPF peptide (**A**) and hydrolysate by rTcCathL1 after 40 min of incubation (**B**), where (1), (2), (3), (4), (5), and (6) are the LPYPQPQ, LQPFPQPQ, LPYPQPQPF, LPYPQPQLPYPQPQ, LQPFPQPQLPYPQPQ, and LPYPQPQLPYPQPQPF peptides, respectively.

**Table 1 ijms-23-07001-t001:** Major products of the hydrolysis of the 33-mer peptide by rTcCathL1.

Peptide	*m*/*z* Value
LPYPQPQ	842.5
LQPFPQPQ	954.5
LPYPQPQPF	1086.6
LPYPQPQLPYPQPQ	1666.9
LQPFPQPQLPYPQPQ	1778.9
LPYPQPQLPYPQPQPF	1911.0

## Data Availability

Not applicable.

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
