# Peer review of "Recombinant Cathepsin L of *Tribolium castaneum* and Its Potential in the Hydrolysis of Immunogenic Gliadin Peptides"

_ijms, 2022, doi:10.3390/ijms23137001_

Round 1
Reviewer 1 Report
The submitted manuscript deals with the characterization of T. castaneum cathepsin L and the potential hydrolysis of immunogenic gliadin peptides.
However, there are some issues that should be addressed:
ABSTRACT
In its current form is too general. Therefore, it is suggested to include the findings presented in the Results section such as stated in lines 184-185
INTRODUCTION
This section includes some very general statements that sometimes are also misleading. Moreover, the cited references are not always appropriate in that context.
For example, the first paragraph is too general and in its current form also misleading. The cited reference (Ref 1) refers to a MEROPS database and probably a review article or the canonical book “Proteolytic enzymes” will be more suitable in this context.
- The statement in lines 39-41 is misleading and irrelevant in this context (see Ref 3)
- The statement in lines 56-60 is too long and hard to understand in its current form.
- Similarly applies to the statement in lines 72-74.
- The authors should confirm and update accordingly, the consecutive numbering of the references. Now Refs 27-35 are cited in the context of Section 4. Materials and Methods, so it is a gap between Section 1. Introduction (where the last citation is Ref 26) and Section 2. Results which starts at Ref. 36.
RESULTS
- The term “created” in line 86 is redundant and could be omitted.
- Similarly, the term “liquid” in line 90 is redundant and could be omitted, as well.
- Why the dialysis step was carried out against distilled water and not buffer where the enzyme will be more stable? (see line 92)
- In line 93 the authors stated the obtained yield of the purified protein; however, no purification Table was included.
- Item 2.3 is written in detail, although it is well known for cathepsin L from other organisms.
- Item 2.7. in line 214 states Fig.7 but the authors should confirm if refers to Fig.8 and update it, accordingly.
DISCUSSION
-Similar to the Introduction, this section includes some very general statements that sometimes are also misleading such as lines 279-282, 289-294, 299-301, among others.
Namely, “…the only possible cure for celiac disease is a lifelong gluten-free diet…”. Well, the above it is not a cure but rather a disease management.
-In line 339 the authors used the term “authenticity” and should confirm whether it refers to “identity”
REFERENCES
Some important references are missing such as:
Kiyosaki T et al. Gliadain, a gibberellin-inducible cysteine proteinase occurring in germinating seeds of wheat, Triticum aestivum L., specifically digests gliadin and is regulated by intrinsic cystatins. FEBS J. 2007 Apr;274(8):1908-17.
Martinez M et al. Plant Proteases: From Key Enzymes in Germination to Allies for Fighting Human Gluten-Related Disorders. Front Plant Sci. 2019;10:721. doi:10.3389/fpls.2019.00721
Author Response
Dear Reviewer 1,
Please see the attachment.

Reviewer 2 Report
The manuscript is very well written and provides a crucial information about possible a non-dietary therapy for celiac disease. I appreciate the authors for this work. I have few minor comments only.
Comment#1: The authors have discussed only 1-2 lined about celiac disease (Introduction and discussion). I suggest writing 1-2 more lines telling what celiac disease is. It will increase the impact of the work.
Comment#2: Please mention the name of the software used to perform statistical tests.
Author Response
Dear Reviewer 2,
Please see the attachment.

Round 2
Reviewer 1 Report
The authors properly addressed all comments raised by the reviewer. The revised version of the manuscript includes all suggestions; thus significantly improving its quality and readability.